# *Hydrocharis laevigata* in Europe

**DOI:** 10.3390/plants12040701

**Published:** 2023-02-04

**Authors:** Pablo Garcia-Murillo

**Affiliations:** Department of Plant Biology and Ecology, Faculty of Pharmacy, University of Seville, 41012 Seville, Spain; pgarcia@us.es

**Keywords:** aquatic plants, spongeplant, AIS, biological invasions

## Abstract

*Hydrocharis laevigata* (Humb. & Bonpl. ex Willd.) Byng & Christenh. [= *Limnobium laevigatum* (Humb. & Bonpl. ex Willd.) Heine], Hydrocharitaceae, is a floating-leaf aquatic plant that is native to inland South America. It is an invasive species in several parts of the world. Reports of its presence in Europe have been recently published: naturalised populations occur in three locations on the Iberian Peninsula. The literature also contains records of the species in Hungary and Poland. In addition, it has been observed in Sweden, Belgium, and the Netherlands. *H. laevigata* is highly adaptable and can profoundly transform habitat conditions in its invasive range, causing major issues for ecosystem conservation and human activities. Until recently, *H. laevigata* was not to be found in natural environments in Europe. Factors explaining its spread include its use as an ornamental plant, the eutrophication of inland waters, and the effects of global warming. With a focus on Europe, this short communication provides information on the species’ distribution, taxonomy, biology, habitat, and negative impacts.

## 1. Introduction

The major decline in freshwater species biodiversity has become one of the most important alterations in ecosystems around the world [1,2]. Pollution, water flow restrictions, and climate change are transforming conditions in freshwater habitats, favouring the establishment of invasive species [3,4,5].

There is a common consensus, supported by the Red List of European Habitats [6], that the key threats to European aquatic and wetland habitats are alterations to hydrological systems, climate change, pollution, and invasive species [7]. In Europe, native freshwater biota are facing increasing competitive pressure due to the arrival and spread of floating-leaf aquatic plants (e.g., *Azolla filiculoides* and *Eichhornia crassipes*) in inland water bodies [8,9,10,11,12,13,14,15]. Of particular concern is the recent appearance of an aggressive invasive alien species (AIS), *Hydrocharis laevigata*, whose common names include South American spongeplant, smooth frogbit, or Amazon frogbit. Newly established in a range of European ponds, rivers, and reservoirs, the species serves as yet another warning about the deterioration of Europe’s freshwater habitats. Its invasive dynamics are reminiscent of those of *Eichhornia crassipes*, including its striking colonisation abilities. This short communication describes *H. laevigata*’s history in Europe, including its emergence in distinct continental regions. It also provides key information about the species’ taxonomy, biology, habitat, and negative impacts. To this end, it will facilitate prevention efforts aimed at invasive species, help control the spread of *H. laevigata*, and bolster the preservation of Europe’s native biodiversity.

## 2. Results

### 2.1. Taxonomy and Nomenclature

*Hydrocharis laevigata* (Humb. & Bonpl. ex Willd.) Byng & Christenh. belongs to the family Hydrocharitaceae (superorder Lilianae). This taxon of monocots contains genera whose members are among the world’s worst aquatic invasive plants, such as *Eichhornia* Kunth and *Hydrilla* Rich.

Phylogenomic, morphological, and anatomical evidence strongly suggest a new vision of the genus *Hydrocharis* L. [16,17], one that includes the species of *Limnobium* Rich., usually considered to be a separate genus by World Flora Online (WFO) [18]. Assuming the species’ reclassification, there are currently five known members of the genus *Hydrocharis* L. Three are spread throughout Afro-Asia, and the other two are found from Mexico to northern Argentina and in the Caribbean. Those previously classified in *Limonium* are native to the Americas and utilise wind pollination. The original members of *Hydrocharis* are autochthonous from the Old World and are insect pollinated [19]. Those differences aside, all five species have a close phylogenetic relationship and display similarities in their vegetative traits [19,20].

*Hydrocharis laevigata* (Humb. & Bonpl. ex Willd.) Byng & Christenh., Global Fl., 4: 53. 2018.

≡ *Salvinia laevigata* Humb. & Bonpl. ex Willd., Sp. Pl., ed. 4, 5: 537. 1810 (basionym).

≡ *Limnobium laevigatum* (Humb. & Bonpl. ex Willd.) Heine in Adansonia, 8: 315. 1968.

≡ *Hydromystria laevigata* (Humb. & Bonpl. ex Willd.) Hunz., Lorentzia, 4: 5. 1981.

≡ *Limnobium spongia* subsp. *laevigatum* (Humb. & Bonpl. ex Willd.) Lowden, Rhodora, 94: 129. 1992).

Type (lectotype designated by [21] (p. 220): Colombia; Bogotá. Humboldt & Bonpland (holotype: B-Willd. 20251).

= *Hydromystria stolonifera* G. Mey., Prim. Fl. Esseq. 153. 1818.

≡ *Limnobium stoloniferum* (G. Mey.) Griseb., Fl. Brit. W. I. 506. 1864.

This species has traditionally been included in the genus *Limnobioum* Rich., under the name *Limnobium laevigatum* (Humb. & Bonpl. ex Willd.) Heine. Other authors have considered this taxon to be a subspecies of *L. spongia* (Bosc) Steud.

### 2.2. Species Description

According to [21,22,23,24], *H. laevigata* is an aquatic stoloniferous plant whose stems are floating on or suspended in the water. They develop floating or emergent leaf rosettes, which are sometimes rooted. The species’ differentiated leaves are petiolate, laminate, cordate to reniform, or orbiculate, stipulate, and dimorphic. The aerial leaves have flat, leathery blades and long erect petioles measuring up to 40 cm; the floating leaves have thick spongy tissue on the lower surface and shorter petioles (minimum of 12 cm). Flowers are about 15 mm wide, unisexual, pedicelled, and enclosed in a spathe. Staminate flowers often occur in clusters of two or three; have three (rarely four) sepals and petals each; possess a variable number of stamens, whose filaments form a single column; and usually exhibit staminodes. Pistillate flowers are solitary; have rudimentary sepals and petals, although the latter may be lacking; and possess a unilocular ovary with six carpels, six styles, and divided linear stigmas that are longer than the perianth segments. Furthermore, female flowers may sometimes exhibit staminodes. The species’ fruit consists of globose or ellipsoid fleshy berry-like capsules (~1.5 cm long and 0.4–3.5 mm wide) that contain as many as 100 ellipsoid seeds measuring around 10 mm long.

### 2.3. Habitat

Found in the tropics and subtropics [22,25], *H. laevigata* grows in a variety of freshwater habitats, such as shallow ponds; lakes; dams; reservoirs; pools; and the margins of slow shady rivers, canals, and ditches; environments with a high degree of eutrophication are particularly hospitable [26,27]. The species occurs at elevations from sea level all the way up to 2800 m. It is absent from fast-flowing watercourses [21,22,26,28], and cold temperatures limit its distribution [25].

### 2.4. Reproduction

*Hydrocharis laevigata* reproduces vegetatively via the fragmentation of the stolon segments connecting the leaf rosettes. The floating rosettes send out stolons bearing ramets on their ends [29]. The rosettes are easily dislodged by waterfowl or large mammals in the species’ native range [26], as well as by water currents or other types of water movement in general. Consequently, *H. laevigata* is able to rapidly and efficiently colonise suitable new habitats [28].

With regards to sexual reproduction, the species mainly utilises wind pollination [17,22,28]. However, water pollination has also been observed, as has insect pollination, mediated by tiny aphid nymphs (Aphididae) and the aphids’ predators, ladybird beetles (Coccinellidae) [24].

After pollination, the pedicel of female flowers tips downwards, such that the fruit develops submerged in water; once it reaches maturity, it breaks open, releasing a mucilaginous mass of about 100 seeds [21].

### 2.5. Global Distribution

The native range of *H. laevigata* extends across the tropics of Central and South America. Naturalised populations can now be found on all continents, with the exception of Antarctica.

According to [22,24,30], this species is naturally found in water bodies in parts of the Americas with tropical, subtropical, and, sometimes, temperate climates. More specifically, these locations are the CARIBBEAN: Antigua and Barbuda, Cuba, the Dominican Republic, Guadeloupe, St. Lucia, Montserrat, Martinique, Trinidad and Tobago, and Puerto Rico; CENTRAL AMERICA: Mexico, Costa Rica, Guatemala, Nicaragua, Panama, and El Salvador; and SOUTH AMERICA: French Guiana, Guyana, Suriname, Venezuela, Brazil, Colombia, Ecuador, Peru, Argentina, Paraguay, and Uruguay. It should also be noted that there may exist some problematic records for South America. For example, although [31] stated that the species is native to the Southern Cone, this assertion is questionable in the case of Chile, as [26] indicates that the country’s oldest herbarium specimen dates back to 1954. The same is likely true for Argentina: while the species’ native range is thought to extend across most of the country, it certainly appears to be invasive in the southernmost regions of Argentina, where it has recently appeared for the first time [32].

As noted in [25,29,33,34,35,36], *H. laevigata* has become naturalised and spread across ASIA: Japan, Taiwan, and Indonesia; AFRICA: Zambia and Zimbabwe; AUSTRALIA: New South Wales and Queensland; NORTH AMERICA: southern Canada as well as the north-eastern and western US; and EUROPE.

### 2.6. Populations in Europe

Based on the scientific literature and the GBIF database, *H. laevigata* is known to occur in six European countries: Belgium, Hungary, The Netherlands, Poland, Spain, and Sweden.

The first record of its presence (as *Limnobium laevigatum*) was in hot springs located in Komárom-Esztergom County, northern Hungary [37]. In September 2018, the species was observed in the area of Fényes springs, north of Tata. At this location, the plants were mainly in a vegetative state, although some did bear flowers. One month later, the plants had spread to the Ferencmajori fishponds, traveling a distance of about 1.3 km. In October 2018, the species turned up at another site within the same county: Dunaalmás, a stream fed by warm water [37]. It was observed that the *H. laevigata* populations became established near hot springs (temperatures = 21–23 °C), which do not freeze in the winter. Recently, Dr. Riezing (pers. comm.) has indicated that these populations in northern Hungary have persisted, likely because of mild winter conditions and the warm water temperatures. However, the plants occur in small groups.

On the Iberian Peninsula, the species seems to have rapidly expanded its range. Since 2018, it has been found at three locations that are quite distant from each other. The first published record for this species (as *Limnobioum laevigatum*) was associated with a reservoir near Madrid [38]. A few years later, it was observed in a stream next to a dam near Córdoba [39]. Finally, it was also reported to occur in the Guadaira, a tributary of the Gualdalquivir, near Seville [40] (Figure 1). There are no indications that *H. laevigata* has disappeared from any of these locations in Spain.

In 2020, *H. laevigata* was observed in southern Poland, in an artificial pond near Agatowa Street in the district of Bieżanów-Prokocim in eastern Kraków [15]. It was noted [15] that the species (as *L. laevigatum*) is likely a sporadic invader in Poland, given that other similar invasive species display constrained ranges due to the country’s cold winter temperatures. Dr. Pliszko (pers. comm.) recently confirmed that this population has disappeared because of the species’ vulnerability to low temperatures. It cannot survive winter conditions in Poland, outside of the habitat found in greenhouses or residences (Figure 1).

The GBIF lists nine geographical records for *H. laevigata* in Europe [41], which include Ganshoren, Belgium; the outskirts of Stockholm, Sweden; and Aalsmer and Bleiswijk in the Netherlands. Only in the latter country are the records associated with herbarium specimens. In the other cases, there are only photographs, no herbarium samples (Appendix A and Appendix B). Two of these locations represent artificial habitats: a greenhouse in Aalsmeer in the Netherlands and the Bergianska Botanick Garden in Sweden. The remaining records for Belgium and the Netherlands were one-time events. Consequently, they can be viewed as sporadic occurrences that have not resulted in *H. laevigata* becoming naturalised. The same is true for the record in Poland. Therefore, at present, the only naturalised *H. laevigata* populations in Europe are in Hungary and Spain (Figure 2).

### 2.7. Invasiveness

As noted in the introduction, *H. laevigata* displays invasive dynamics reminiscent of those of *Eichhornia crassipes*, including its marked capacity for colonisation [25,32,42]. Indeed, like other invasive aquatic plant species, it is fast growing and thrives in still waters containing high concentrations of nutrients [43]. Under these conditions, it can outcompete native aquatic plant species and profoundly transforms the habitats it invades. While the impacts of this species have yet to be fully described, past observations [22,25,28,44] indicate that the floating rosettes block light from reaching the water below, indirectly fostering anoxic conditions. *H. laevigata* can thus reduce the abundance and diversity of aquatic plants and animals (e.g., fish, phytoplankton, submersed plants) by rendering water bodies unsuitable for other organisms. The species can also generate mats that cause damage to human infrastructures (e.g., by clogging irrigation pipes or sewers, by damaging pumps or generators in hydroelectric power stations). Consequently, *H. laevigata* is considered to be one of the most harmful weeds in the United States and Australia, given its high degree of invasiveness combined with its negative impacts on water quality and aquatic biodiversity [25,29,42,44].

While *H. spongia* (= *Limnobium spongia*) is rarely cultivated, *H. laevigatum* is commonly found in aquaria and garden ponds, largely because of its ecological breadth and pronounced reproductive capacity [32,35,36,37,45].

### 2.8. Control

Typical methods for controlling *H. laevigata* are mechanical harvesting and herbicide treatments [45,46]. In the Sacramento-San Joaquin Delta, diquat dibromide has been used, either on its own or in tandem with low levels of glyphosate [46]; this work yielded unclear results. Another study [47] evaluated the performance of the herbicides imazamox, penoxsulam and topramezone. Imazamox (with 1% methylated seed oil surfactant) was found to control all the growth stages of *H. laevigata*; penoxsulam was particularly effective in targeting seedling and rosette growth. Research is currently lacking on potential biological control methods. However, [36] points out that certain North American herbivores display a dietary specialisation for American frogbit (*H. spongia*). They might therefore be candidate biological control agents.

## 3. Discussion

In summary, *H. laevigata* is an invasive plant species that poses a significant threat to ecosystems and certain human activities in its non-native range. Its likelihood of establishment is considerably elevated in nutrient-rich, slow-moving freshwater systems subject to mild winter temperatures [25,26,27,28], such as those found in southern Europe or the northern Mediterranean. Its invasion dynamics are similar to those of *Eichhornia crassipes* [25,32,42].

As indicated earlier, the ornamental plant trade is the main source of *H. laevigata* (and *Eichhornia crassipes*) introductions. This fact explains how the species has spread so far from its native range to parts of the world as diverse as Australia, Zimbabwe, Japan, and Spain.

The range expansion of *H. laevigata* within Europe underscores the deterioration of the continent’s inland aquatic ecosystems. Such is, above all, the result of eutrophication, but other causal factors include changes to hydrological regimes (i.e., reduced levels of water flow), decreasing degrees of water transparency, and global warming, which are simultaneously transforming natural aquatic environments [5,48,49,50,51,52,53]. These ecological shifts are harmful to native aquatic plant populations while promoting the establishment of invasive species such as *H. laevigata*, leading to profound transformations in European ecosystems [12,14,54,55,56,57].

It is curious to note that, while the invasive *H. laevigata* has begun to expand its range in Europe, the native *H. morsus-ranae*, a close relative, is experiencing decline. While the latter has always been patchily distributed across western Europe, populations are shrinking in certain regions [58]. For example, on the Iberian Peninsula, *H. morsus-ranae* now occurs in a single area [59], a range constriction that took place over the last century. Probable causes include habitat degradation due to farming and industrial activities as well as changes in the hydrochemical characteristics and hydrological regimes of water bodies [58]. At the same time, *H. morsus-ranae* has become an AIS in the north-eastern USA and southern Canada [60,61].

Hence, given the plant’s recent range expansion in Europe, it is crucial to maximise efforts to rapidly detect its presence. By acting now, during the early stages of the invasion, it is more likely that *H. laevigata* can be controlled [62,63]. At particular risk is the Mediterranean, where conditions in rivers, streams, lakes, ponds, reservoirs, and canals favour the arrival and establishment of *H. laevigata*.

In this regard, it should be added that the introduction and spread of IAS are a major concern for the European Union; for those reasons, Regulation (EU) No 1143/2014 of the European Parliament and of the Council of 22 October 2014 on the prevention and management of the introduction and spread of invasive alien species was published, which is being developed and updated through different implementing regulations and which establishes rules to prevent, minimize, and mitigate the adverse effects of IAS. This regulation is associated with a list of IAS: “the Union list”, which is the fundamental instrument for implementing the aforementioned regulation. *Hydrocharis laevigata* is not included in this list. Taking into account the problems that this species causes to native biodiversity and to human activities in other parts of the world, it is hoped that the information contained in this brief note will serve as a starting point to initiate actions to include *Hydrocharis laevigata* in the “the Union list”, something that will undoubtedly benefit the conservation of European biodiversity and reduce its harmful effects on human activities.

## 4. Materials and Methods

This study was based on an extensive literature analysis, data from field surveys, and the examination of herbarium specimens (at MA, SEV, COFC; see Appendix A).

The Geospatial Conservation Assessment Tool (http://geocat.kew.org/, accessed on 1 November 2022) was used to create the map (Figure 2) showing where *H. laevigata* has been observed (location descriptions: Appendix A and Appendix B).

## Figures and Tables

**Figure 1 plants-12-00701-f001:**
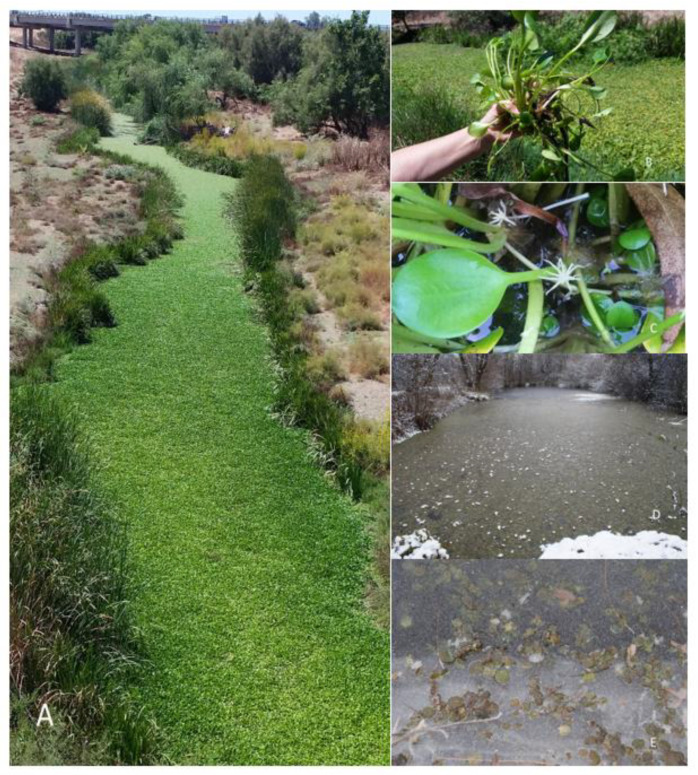
(**A**,**B**) *Hydrocharis laevigata* habitat in the outskirts of Seville, Spain. (**C**) Male flowers, photos taken by P. Garcia Murillo in summer 2021; (**D**,**E**) habitat in the outskirts of Krakow, Poland, photos taken in November and December 2020, courtesy of Dr. Arthur Pliszko.

**Figure 2 plants-12-00701-f002:**
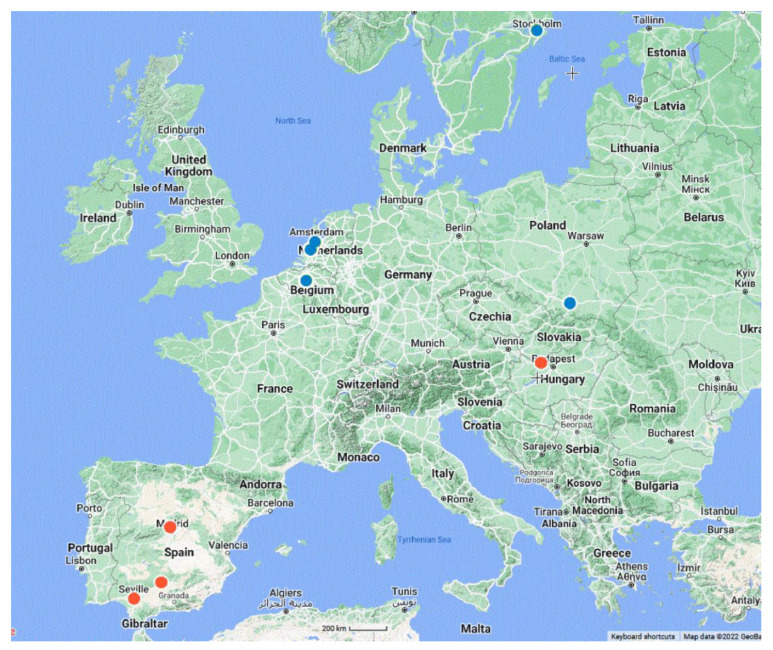
Locations where *Hydrocharis laevigata* has been observed in Europe. Blue dots: sporadic presence. Red dots: naturalised populations. When the name of the city has a different spelling of the English alphabet, it appears below in its original spelling. Map created using The Geospatial Conservation Assessment Tool.

## Data Availability

Not applicable.

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
