# Peer review of "Hydrocharis laevigata in Europe"

_plants, 2023, doi:10.3390/plants12040701_

Round 1

Reviewer 1 Report

I have read the ms "Hydrocharis laevigata (Humb. & Bonpl. ex Willd.) Byng & Christenh. in Europe" with great interest. The writing is good and clear (except for a few comments given at the end), and I found the information given there well organized and important for everyone dealing with invasive species. However, I found very little difference between the data reported in the manuscript and the data reported on the CABI representative web page, "https://www.cabidigitallibrary.org/doi/10.1079/cabicompendium.115273". 

Thus, I see no clear reason why this kind of manuscript might be published in a high impact journal like "Plants." The only way to reconsider its publication is to conduct a proper "Risk Screening" exercise for Europe and provide evidence of its potential impact, e.g., under predicted climate change scenario. 

Thus, with regret, I cannot recommend this manuscript for publication in its current form and only if it can be significantly improved by means of risk screening protocolo (a.g. ISK family of tools) that also includes adding the methodological section, than it might worse to be re-considered.

Figure 1 - looks strange. Should be re-arranged.

Figure 2 - Green/blue dots are not shown. Instead red/blue dots are provided and should be corrected

Reviewer 2 Report

Just few comments about this interesting and useful note.

In the Discussion, it could be useful to discuss the EU Regulation 1143/2014 on Invasive Alien Species and the possibility/need of updating the list of invasive alien species of Union concern.

Line 65: laevigatum (in italics)

Line 70: I think Limnobium stoloniferum and Hydromystria stolonifera are homotypic synonyms, so in line 70 the symbol "=" has to be changed with "≡", and "(Meyer)" with "(G. Mey.")

Round 2

Reviewer 1 Report

After the rebuttals given by the author to the main concerns in the reviews, I am convinced that the manuscript is worthy of publication. Since it is in good shape now, I suggest to accept as it is.